# Replacement of Meat with Non-Meat Protein Sources: A Review of the Drivers and Inhibitors in Developed Countries

**DOI:** 10.3390/nu13103602

**Published:** 2021-10-14

**Authors:** Marion R. Eckl, Sander Biesbroek, Pieter van’t Veer, Johanna M. Geleijnse

**Affiliations:** Division of Human Nutrition and Health, Wageningen University, Stippeneng 4, 6708 WE Wageningen, The Netherlands; sander.biesbroek@wur.nl (S.B.); pieter.vantveer@wur.nl (P.v.V.); marianne.geleijnse@wur.nl (J.M.G.)

**Keywords:** meat replacement, non-meat protein source, environmental sustainability, consumer preference, food choice

## Abstract

The overconsumption of meat has been charged with contributing to poor health and environmental degradation. Replacing meat with non-meat protein sources is one strategy advocated to reduce meat intake. This narrative review aims to identify the drivers and inhibitors underlying replacing meat with non-meat protein sources in omnivores and flexitarians in developed countries. A systematic search was conducted in Scopus and Web of Science until April 2021. In total, twenty-three studies were included in this review examining personal, socio-cultural, and external factors. Factors including female gender, information on health and the environment, and lower price may act as drivers to replacing meat with non-meat protein sources. Factors including male gender, meat attachment, food neophobia, and lower situational appropriateness of consuming non-meat protein sources may act as inhibitors. Research is needed to establish the relevance of socioeconomic status, race, ethnicity, religion, health status, food environment, and cooking skills. Future studies should prioritize standardizing the definitions of meat and non-meat protein replacements and examining factors across different consumer segments and types of non-meat protein sources. Thereby, the factors determining the replacement of meat with non-meat protein sources can be better elucidated, thus, facilitating the transition to a healthier and more sustainable diet.

## 1. Introduction

Over the years, there has been an increasing body of research advocating for a reduction in the overconsumption of meat in order to mitigate negative health consequences and environmental burdens [1,2]. Despite being a valuable source of nutrients including protein, vitamin B-12, iron, and zinc [3], red and processed meat, in particular, have been shown to be associated with an increased risk of cardiovascular disease, stroke, cancer, as well as total mortality [3,4,5]. Plant-based diets have positive health benefits including a reduced risk of type II diabetes mellitus and cardiovascular disease [6,7,8]. Going further, meat production has been charged with contributing to environmental degradation including increased greenhouse gas emissions, loss of biodiversity, and disturbances in nitrogen-phosphorus soil balance [9,10,11]. By 2050, the world’s population is expected to increase from 8 to 10 billion people [12]. Combined with continued global warming, such population growth will necessitate a further increase in food production, thereby, exacerbating the burden of non-communicable diseases and devastation of the environment [1]. As such, decreasing meat consumption in overconsuming developed countries remains key to abating such disastrous consequences in the coming years.

Overall, strategies to decrease meat consumption exist on a continuum from reduction to elimination [13]. Reduction strategies include decreasing the amount of meat consumed and often increasing the proportion of other non-meat foods at mealtimes (e.g., vegetables) [13,14]. Replacement strategies include either partially substituting meat with non-meat protein sources in traditional meat-based recipes (e.g., replacing a portion of beef with mushrooms in hamburgers) or fully substituting meat with non-meat protein sources (e.g., replacing pork with black beans in tacos) [13,14]. Decreasing the portion size of meat may be more feasible for many consumers as it does not require any alteration of the meal recipe and context or procurement of new cooking skills. In contrast, substituting meat with a non-meat protein source may be more or less feasible depending upon the degree of substitution, type of non-meat protein source, and necessary cooking skills involved to implement the recipe. Besides the sensory pleasure derived from meat, meat continues to maintain a strong symbolic place in many Western cultures often dominating the meal context as the central food item emblematic of higher socioeconomic status and masculinity [15,16]. Consequently, reducing meat consumption regardless of the strategy employed remains a challenge for many consumers.

With this in mind, it is crucial that we elucidate the underlying drivers and inhibitors to reducing meat consumption and particularly replacing meat with non-meat protein sources for public health policy, food industry, and dietitians and other health professionals in order to best facilitate a timely transition to a healthier and more sustainable global food system. To date, many reviews have looked at the factors involved in consumers reducing meat consumption in general but have not specifically examined the factors involved in consumers replacing meat with non-meat protein sources [17,18,19,20,21]. Although important fixtures in transforming the global food system, vegetarians and vegans constitute a small percentage of the population [22,23,24,25], and strict elimination of meat may not be realistic or necessary for most consumers particularly as a first step to reducing meat intake [26]. Consumer segments including omnivores, often referred to as meat-eaters, and flexitarians, often referred to as meat-reducers, comprise a much larger percentage of the population in many developed countries [22,23,26]. Therefore, understanding the motivations of omnivores and flexitarians is key to enacting a sizeable and long-term shift towards consuming less meat in developed countries.

As such, the aim of this review is to identify the drivers and inhibitors underlying consumer behavior of replacing meat with non-meat protein sources in omnivores and flexitarians in developed countries. In this way, we can contribute to better elucidating the motivations, attitudes, and behavior of omnivores and flexitarians in order to assist future research investigating the transition to and ultimately acceptance of healthier and more sustainable protein sources in society.

## 2. Materials and Methods

We conducted a comprehensive literature search in Scopus and Web of Science (Core Collection) in order to identify all studies examining the drivers and inhibitors to replacing meat with non-meat protein sources. The timespan of the search extended from the earliest date available in the databases up to April 2021. The search strings for the respective databases consisted of keywords relating to the replacement of meat with various non-meat protein sources including plant-based and alternative protein sources (Appendix A and Appendix B). The initial search was supplemented by a manual search of reference lists of relevant articles to identify studies not retrieved in either Scopus or Web of Science.

Only studies consisting of human consumers that eat meat ((i.e., consumers described as omnivores (or meat-eaters) or flexitarians (or meat-reducers)) from developed countries were included in this review. Studies consisting only of consumers described as pescatarians, vegetarians, or vegans or from developing or transition countries were excluded. Additionally, only studies that utilized a non-meat protein source to replace meat and that examined the drivers and inhibitors relating to the perception, awareness, attitude, motivation, willingness, and behavior to replace meat with a non-meat protein source were included. Figure 1 describes the literature search and provides details for the reasons for excluding studies including: (1) irrelevant topic; (2) irrelevant population; (3) irrelevant exposure; (4) irrelevant outcome; (5) irrelevant study design; (6) no full-text available; and (7) no English translation available.

The title and abstract of articles were first screened for inclusion and exclusion criteria, and when these criteria were met, the full texts of the articles were retrieved and screened for these criteria. From the selected articles, we extracted data on the authors and year of publication; study location and design; population characteristics; data collection; non-meat protein replacements; explanatory and dependent variables; as well as the outcomes pertaining to the replacement of meat with non-meat protein sources. One researcher was involved in screening articles for inclusion and exclusion criteria and data extraction. A second researcher randomly cross-checked the screening of articles and data extraction and discussed any uncertainties and disagreements with the first researcher.

In this review, we utilized the theoretical framework of factors that influence meat-eating behavior by Stoll-Kleemann and Schmidt (2017) to organize and summarize our findings in the following sections with minor alterations (Figure 2) [17]. This framework was chosen as it provides a comprehensive overview of the personal, socio-cultural, and external factors that influence consumers’ meat-eating behavior [17]. This framework considers internal and external incentives related to reducing meat consumption and furthermore the interrelationships among these factors [17]. This framework is based on the pro-environmental model developed by Kollmuss and Agyeman (2010), which also asserts the complexity and synergism of internal and external factors in determining individuals’ propensity to partake in pro-environmental behavior that seeks to mitigate the negative impact of an individual’s behavior on the environment [27].

## 3. Results

### 3.1. Study Characteristics

In total, twenty-three studies were included in this review. Table 1 provides an overview of the characteristics of these studies. The studies were published from 2011 [28] to 2021 [14,29,30,31,32,33,34,35,36]. Of the twenty-three studies, seventeen were conducted in Europe including [28,29,30,31,32,33,34,36,37,38,39,40,41,42,43,44,45]: Belgium [34,44], Denmark [29], Finland [29], France [31,36], Germany [29,32,33,34,36,41,45], Hungary [40], Iceland [29], Italy [39,43], The Netherlands [28,30,42], Romania [29], and the United Kingdom (UK) [36,37,38]. Five studies were conducted in North America [13,35,46,47,48] with four studies coming from the United States (US) [13,35,47,48] and one from Canada [46]. One study was conducted in New Zealand [14]. Importantly, three studies were conducted with consumers of multiple countries including: Denmark, Finland, Germany, Iceland, and Romania [29]; Germany, France, and the UK [36]; and Germany and Belgium [34]. Most studies (*n* = 20) employed quantitative research methods [13,28,29,30,31,33,34,35,36,37,38,39,40,41,42,43,44,45,47,48] with nine studies utilizing surveys [13,29,33,36,40,41,43,44,45] and eleven studies an experimental design [28,30,31,34,35,37,38,39,42,47,48]. One study was considered a qualitative study and utilized semi-structured interviews [14]. Two studies employed a mixed-methods approach of quantitative and qualitative research methods in their study design [32,46]. In terms of replacements of meat, seventeen studies investigated full replacements of meat with non-meat protein sources [14,28,29,30,31,32,36,37,38,39,40,41,42,43,44,45,46], whereas six studies investigated a partial replacement of meat with non-meat protein sources such as mushrooms or legumes [13,33,34,35,47,48]. Most studies (*n* = 19) investigated replacing meat with plant-based protein sources, such as Quorn^®^, tofu, lentils, or legumes [13,14,28,29,30,31,32,33,34,35,36,37,38,39,41,42,46,47,48]; and two studies each respectively investigated replacing meat with insects [40,44] and cultured meat [43,45]. Table 2 provides a summary of the findings for each of the personal, socio-cultural, and external factors identified among the included studies. 

### 3.2. Personal Factors

#### 3.2.1. Socio-Demographics

##### Age

Six studies examined age as a factor influencing the replacement of meat with non-meat protein sources [13,36,37,38,44,45]. In a discrete choice experiment (DCE) conducted in the UK in 2016, 233 meat-eaters and meat-reducers were segmented into five consumers segments based on their preferences for product-attributes of ground meat and ground meat substitute (i.e., soy, tofu, and Quorn^®^), which varied by age [37]. Organic (79%), green (45%), and taste-driven (46%) consumers were more likely to be between 18–34 years (yr). Price-conscious consumers were more likely to be between the ages of 35–55 yr (65%), and healthy consumers were more likely to be older than 55 yr (79%) [37]. In another DCE conducted in the UK in 2019 with 400 participants, age also varied among the five consumer segments of meat-eaters and meat-reducers based on their preferences for product attributes of ground meat and ground meat substitute (Quorn^®^) [38]. Of the meat-eaters, traditional meat-eaters were more likely to be older and showed a greater preference for ground beef compared to price-conscious meat-eaters who were influenced more by, not just the type, but the price of ground meat or ground meat substitute [38]. In an online survey conducted in Belgium in 2015 with 368 participants, every 10-year increase in age was associated with a 27% reduction in the readiness to adopt insects as a meat substitute [44]. Similarly, in a multi-national online survey conducted in Germany, France, and the UK in 2021 with 1734 participants, it was found that older participants were more likely to provide lower ratings for the expected tastiness, healthiness, and environmental friendliness of pea burgers [36]. Nevertheless, it was not found that age was associated with the expectations for taste, health, and environmental friendliness of algae burgers [36].

Furthermore, a 2020 US survey with 602 participants found no association between age and the assessment and acceptance of blending mushrooms into traditionally meat-based foods to reduce meat consumption [13]. Additionally, a 2020 survey in Germany found that age was not shown to moderate the attitudes pertaining to the intention to try, eat, or promote cultured meat to friends among 713 German participants [45].

##### Gender and Sex

Seven studies investigated the role of gender or sex in the replacement of meat with non-meat protein sources [13,35,36,37,38,44,45]. Similar to age, the five consumer segments of meat-eaters and meat-reducers in the 2016 UK DCE varied by gender [37]. Green (62%) and healthy (78%) consumers tended to identify as female, whereas taste-driven (61%) and organic (83%) consumers tended to identify as male. Price-conscious consumers, on the other hand, largely identified equally as male and female [37]. In the 2019 UK DCE, the five consumer segments of meat-eaters and meat-reducers also varied by gender [38]. Of the meat-eaters, empowered consumers, who were influenced more by the type, production, and fat content of ground meat or ground meat substitute, were more likely to identify as female than the traditional and price-conscious consumers. Additionally, meat-reducers (67%) in general were more likely to identify as female than meat-eaters (54%) with the meat-reducer consumer segments of health curtailers and sustainable consumers consisting of 86% and 70% females respectively [38]. In a 2021 US experiment, the results showed that participants identifying as male preferred the high-meat dishes without partial replacement of meat with legumes and vegetables more than the participants identifying as female [35]. In a Belgian survey, participants identifying as male were more than twice as likely to adopt insects as a substitute for meat compared to participants identifying as female [44].

Contrastingly, in a recent US survey, gender was not associated with the acceptance of blending mushrooms into traditional meat-based foods; however, participants identifying as female assessed the blending concept more favorably than the participants identifying as male [13]. Additionally, a 2020 German survey found that gender was not shown to moderate the attitudes pertaining to the intention to try, eat, or promote cultured meat [45].

In a multi-national survey in Germany, France, and the UK investigating the impact of sex, male participants were associated with providing lower ratings for the expected environmental friendliness of pea burgers [36]. Similar to age, however, sex was not shown to influence the expected tastiness, healthiness, and environmental friendliness of algae burgers [36].

##### Socioeconomic Status

Five studies assessed education and income as factors influencing the replacement of meat with non-meat protein sources [13,37,38,44,45]. Similar to age and gender, the five consumer segments of meat-eaters and meat-reducers varied by income in the 2016 UK DCE [37]. Both price-conscious (66%) and organic (100%) consumers tended to be of lower income [37]. In the 2019 UK DCE, the five consumer segments of meat-eaters and meat-reducers also varied by income [38]. Of the meat-eaters, empowered meat-eaters were found to have a higher proportion of participants earning a higher income [38].

Contrastingly, a recent US survey found that neither income nor education was associated with the assessment or acceptance of blending mushrooms into traditional meat-based foods [13]. In a Belgian survey, education was also not shown to be associated with the readiness to adopt insects as a substitute for meat among participants [44]. Similarly, education was not shown to moderate the attitudes pertaining to the intention to try, eat, or promote cultured meat among participants in a 2020 German survey [45].

#### 3.2.2. Sensory and Hedonic Aspects

Eight studies examined the role of taste, texture, and appearance in the replacement of meat with non-meat protein sources [13,28,31,32,35,44,47,48]. On blind tasting alone, a 2021 experiment in France found that participants preferred to purchase the pork-based sausage over the plant-based sausage [31]. Similarly, a 2021 German survey found that participants perceived all meat products as being tastier than corresponding plant-based meat alternatives (e.g., chicken nuggets versus vegetarian nuggets) [32]. Furthermore, both omnivores and flexitarians in this study rated meat as being expected to perform better than meat alternatives in terms of flavor and texture. In particular, steak was perceived as being the tastiest food product in this survey compared to tofu, chicken nuggets, vegetarian nuggets, wiener sausages, and vegetarian sausages [32]. In a Belgian survey, for every one unit increase in the importance of taste when evaluating meat, there was a 61% decrease in participants’ readiness to adopt insects as a substitute [44]. Interestingly, a 2011 experiment conducted in the Netherlands found that the flavor and texture of specific meat substitutes influenced the perceived appropriateness of a meat substitute in a meal context less than the actual shape of the meat substitute itself—whether in pieces or ground [28]. In this experiment, participants liked Quorn^®^ pieces more than Quorn^®^ ground in the rice and salad dishes [28].

In terms of partial replacements of meat, participants in a recent US survey rated the perceived taste benefits as an intermediate reason for blending mushrooms into traditional meat-based foods falling behind perceived health and cost benefits [13]. A US experiment conducted in 2016 examined participants' preferences for beef carne asada and beef taco recipes in which a portion of the beef had been partially replaced by mushrooms [47]. In this experiment, participants liked the 100% beef carne asada recipe more in terms of flavor and texture compared to the carne asada recipe that replaced 50% of the beef with mushrooms. Contrastingly, there were no differences in the flavor and texture among the 100% beef taco recipes and the mushroom-containing beef taco recipes containing either 50% or 80% mushrooms. Although participants rated the appearance of the 100% beef taco recipe higher than the appearance of the mushroom-containing beef taco recipes, a correlation analysis revealed that flavor was the best predictor for the overall liking of the beef taco recipes followed by texture; but, appearance was not related to the overall liking of the beef taco recipes [47].

Another US experiment conducted in 2018 similarly investigated participants' preferences for pork carnitas arepas and chicken tikka masala in which a portion of the meat had been partially replaced by legumes [48]. Participants liked the high-meat arepas recipe more for flavor, texture, and appearance than the low-meat arepas and the high- and low-meat chicken tikka masala recipes. The high-meat versions of the arepas and chicken tikka masala recipes were also liked more than the low-meat versions of these recipes; however, no differences were found in the texture and appearance of the high- and low-meat versions. Notably, however, spicy versions of the arepas and chicken tikka masala recipes were liked more for flavor and texture than the regular versions across all meat levels [48].

In a 2021 US experiment, there were no differences among the East Asian bowls in terms of flavor regardless of whether meat had been replaced by legumes and vegetables or the spiciness of the dish [35]. Nevertheless, it was found that not having enough flavor complexity resulted in a decrease of 10 on a 100 scale in overall liking across all participants and bowls. For participants who felt their bowl was not spicy enough, there was a decrease of 8 in overall liking. For participants who felt the bowl was too spicy, there was an overall mean liking drop of 15 [35].

#### 3.2.3. Hunger Cues

Three studies investigated the role of hunger and satiety in the replacement of meat with non-meat protein sources [32,35,42]. In a longitudinal experiment conducted in the Netherlands, the effect of repeated exposure to Quorn^®^ or tofu on product liking was investigated [42]. In this experiment, it was found that the hungrier the participant, the more likely they were to like Quorn^®^ or tofu [42]. In a 2021 US experiment, however, there were no differences in participants’ ratings of satiation, or the feeling of fullness, or satisfaction among East Asian bowls regardless of the meat level [35]. In a 2021 German survey, steak was perceived by participants to being the most filling food product compared to tofu, chicken nuggets, vegetarian nuggets, wiener sausages, and vegetarian sausages [32].

#### 3.2.4. Personality Traits

##### Food Neophobia and Food Technology Neophobia

Seven studies assessed food neophobia and food technology neophobia as factors influencing replacement of meat with non-meat protein sources [13,33,34,36,40,42,44]. In two online surveys conducted in Hungary and Belgium, food neophobia, or the propensity to avoid consuming new foods, was identified as a barrier to the readiness of participants to adopt insects as a substitute for meat [40,44]. In a Belgian survey, there was an 84% and 55% decrease in the readiness of participants to adopt insects as a substitute for meat respectively for every one unit increase in food neophobia and food technology neophobia, or the propensity to avoid consuming foods produced by new technologies [44]. In the multi-national survey conducted in Germany, France, and the UK, the more food neophobic the participant, the lower the ratings provided for the expected tastiness, healthiness, and environmental friendliness of pea and algae burgers [36]. In a 2021 German survey, it was found that the more food neophobic the participant, the less likely they were to choose the meat-hybrid option consisting of 40% plant-based protein [33]. Similarly, in a recent US survey, food innovativeness, or being open to using new foods or ingredients, was associated with a more positive assessment of blending mushrooms into traditional meat-based foods, which was associated with a greater acceptance of blending [13]. While food innovativeness was associated with a more positive assessment of blending for all participants, it was found to have a greater influence on regular consumers who consumed the same or an increased amount of red meat in contrast to transitional consumers either having reduced or at least considered reducing their red meat consumption [13].

Nevertheless, in a 2021 DCE conducted in Germany and Belgium, food neophobia did not affect the overall product liking of meat-hybrids consisting of 50% or 80% plant-based protein [34]. Additionally, food neophobia did not have an effect on the product liking of Quorn^®^ or tofu in the longitudinal experiment in the Netherlands [42].

##### Variety-Seeking

One study assessed variety-seeking personality traits as a factor influencing the replacement of meat with non-meat protein sources [42]. In the longitudinal experiment conducted in the Netherlands, variety seeking, or the tendency of consumers to switch between food products to prevent boredom, had an effect on the product liking of Quorn^®^ or tofu only in interaction with product and time [42]. The greater the number of different meals used by the participants, the less likely they were to like Quorn^®^ or tofu [42].

#### 3.2.5. Knowledge and Skills

##### Information on Health and the Environment

Three studies examined the role of information on health and the environment in the replacement of meat with non-meat protein sources [31,39,43]. In a theoretical willingness to pay (WTP) experiment in Italy in which 119 participants were asked to indicate the amount of money they were willing to pay for beef and soy burgers, successive rounds of explanatory messaging on health and the environment resulted in a relative decrease of −1.6% in the WTP for beef burgers and relative increase of +3.6% in the WTP for soy burgers [39]. Additionally, successive rounds of explanatory messaging on health and the environment resulted in a relative decrease of −23.0% in the chosen quantities of the beef burger and a relative increase of +45.6% in the chosen quantities of soy burgers [39]. In an experiment in France, no difference was found in the willingness to purchase the plant-based sausage after the first message on either health or the environment [31]. After the second message on health or the environment, however, there was an increase in the willingness to purchase the plant-based sausage [31]. In an Italian survey assessing the impact of information on the willingness to try, buy, or purchase cultured meat in Italy, participants showed better agreement with the information provided on the extrinsic attributes of cultured meat, such as its impact on sustainability, security, and animal welfare, in contrast to information provided on the intrinsic attributes of cultured meat, such as the laboratory production and flavor and nutrients [43].

##### Cooking Skills and Food Knowledge

Four studies investigated the role of cooking skills and food knowledge in the replacement of meat with non-meat protein sources [13,14,44,45]. In a qualitative study conducted in New Zealand with 23 young adults, it was found that participants who described themselves as being more confident and experienced in cooking substituted meat with plant-based proteins such as legumes, lentils, and tofu [14]. Conversely, participants who described themselves as being less confident and less experienced in cooking preferred to substitute meat with more convenience-oriented, plant-based proteins such as vegetarian patties and sausages [14]. In a recent US survey, food knowledge, or the knowledge of foods and cooking, was not associated with a positive assessment of blending mushrooms into traditional meat-based foods and, thereby, was not associated with a greater acceptance of this blending concept by participants [13].

In terms of familiarity with alternative protein sources, a 2020 German survey found that pre-knowledge of or familiarity with cultured meat was shown to increase the ethical beliefs of cultured meat but did not impact the emotional objections of cultured meat being unnatural or disgusting or global diffusion optimism of cultured meat being affordable and capable of solving world nutrition problems [45]. In a Belgian survey, participants who claimed to be more familiar with insects were 2.6 times more likely to be ready to adopt insects as a substitute for meat compared to those who claimed to be unfamiliar with eating insects or did not know what eating insects entailed [44].

#### 3.2.6. Emotions and Cognitive Dissonance

One study assessed cognitive dissonance of meat-eating behavior, or the inconsistency between caring for animals as pets yet consuming animals as meat in the diet, as a factor influencing the replacement of meat with non-meat protein sources [41]. In a 2020 German survey, participants were less willing to substitute meat with meat substitutes such as Quorn^®^, tofu, seitan, or soy schnitzel when they scored higher for unapologetic justification strategies to consume meat compared to those that scored lower [41]. The unapologetic justification strategies included: pro-meat attitude favoring a taste for meat; denial of animal suffering; hierarchical justification that humans are superior to animals; religious justification; health justification; human destiny that humans are destined to consume animals; and slaughter justification that denies animal suffering in slaughterhouses [41].

#### 3.2.7. Values and Attitudes

##### Health and Environment

Five studies examined the role of the importance of health and the environment in the replacement of meat with non-meat protein sources [13,32,33,34,44]. In a 2021 German survey, it was found that the higher a participant rated the meat-hybrid in terms of health, the more likely they were to choose the meat-hybrid consisting of 40% plant-based protein compared to the corresponding meat product consisting of 100% meat [33]. While higher ratings for meat-hybrids in terms of the environment and animal welfare were also associated with an increased likelihood of participants choosing the meat-hybrid compared to the 100% meat product, health was found to exert a larger influence on choosing the meat-hybrid than the environment or animal welfare [33]. Similarly, participants ranked perceived health benefits as the top reason but sustainability benefits as the last reason for consuming blended foods in which mushrooms partially replaced a portion of meat in traditional meat-based foods in a recent US survey [13].

In a DCE conducted in Germany and Belgium, participants rated the meat-hybrid consisting of either 50% or 80% of plant-based protein and the 100% plant-based vegetarian alternative as healthier compared to the corresponding 100% meat product [34]. Nevertheless, the same DCE found that the lower the health consciousness of the participant, the lower their preference for meat [34]. Likewise, a Belgian survey found that for every one unit increase in the belief that meat is nutritious and healthy, there was a 64% reduction in the willingness of participants to adopt insects as a substitute for meat [44]. In terms of the environment, however, this same survey showed that for every one unit increase in the attention participants pay to the environmental impact of food, there was a 71% increase in the readiness to adopt insects as a substitute for meat [44]. In a 2021 Germany survey, omnivores perceived meat as performing better for protein content, fat content, and environmental friendliness compared to meat alternatives [32]. Although flexitarians in this study perceived meat as performing better for protein content, they perceived meat substitutes as performing better for fat content and environmental friendliness. Furthermore, participants perceived steak in particular as being the healthiest and protein-rich food item among tofu, chicken nuggets, vegetarian nuggets, wiener sausages, and vegetarian sausages [32].

##### Plant Protein Sources and Production

Three studies investigated the attitudes towards specific protein sources and production methods as factors influencing the replacement of meat with non-meat protein sources [13,29,45]. An online survey was conducted with female participants in Denmark, Finland, Iceland, Germany, and Romania to examine attitudes concerning a meat analogue of wiener sausages containing rapeseed protein [29]. In all countries, attitude towards using plant protein in food production was shown to influence both the intention to substitute meat protein in the diet as well as the attitude towards using rapeseed protein as an ingredient in meat analogues. Furthermore, the attitude towards rapeseed was also shown to influence the attitude towards meat analogues [29]. In a 2020 Germany survey, principal component analysis was utilized to identify three attitudinal dimensions of participants in terms of the intention to try, eat, and promote cultured meat: ethical advantage (e.g., ecological, animal welfare), emotional objections (e.g., unnatural, disgusting), and global diffusion optimism (e.g., affordable, possible global solution) [45]. The ethical beliefs were found to be the primary driver in the intention to try, eat, and promote cultured meat in the future followed by emotional objections and finally global diffusion optimism [45]. In a recent US survey, a positive consumer assessment of blending mushrooms into traditional meat-based dishes was associated with a greater acceptance of blending as a means to reduce meat consumption [13].

##### Vegans and Vegetarians

One study assessed the role of attitudes towards vegans and vegetarians in the replacement of meat with non-meat protein sources [36]. In the multi-national online survey conducted in Germany, France, and the UK, participants who were more negative towards vegan and vegetarian lifestyles provided lower ratings for the expected tastiness, healthiness, and environmental friendliness of pea and algae burgers compared to those who were not negative towards vegan and vegetarian lifestyles [36].

##### Others

In a 2021 Germany survey, participants perceived steak in particular as being the most natural, masculine, and festive among tofu, chicken nuggets, vegetarian nuggets, wiener sausages, and vegetarian sausages [32]. In a recent US survey, participants rated the perceived culinary benefits as the second to last reason preceding perceived sustainability benefits for blending traditional meat-based dishes with mushrooms [13].

#### 3.2.8. Habits

##### Healthy Eating

One study examined healthy eating as a factor influencing the replacement of meat with non-meat protein sources [13]. In a recent US survey, healthy eating was associated with a more positive assessment of blending mushrooms into traditional meat-based dishes [13]. While healthy eating was associated with a more positive assessment of blending for all participants, it was found to have a greater influence on transitional consumers either having reduced or at least considered reducing their red meat consumption in contrast to regular consumers who consumed the same or an increased amount of red meat at the time [13].

##### Consumption of Meat

Six studies investigated the role of consumption of or attachment to meat in the replacement of meat with non-meat protein sources [29,33,34,36,40,44]. In an online survey conducted in Denmark, Finland, Germany, Iceland, and Romania, female participants who consumed less meat were associated with being more likely to purchase meat analogues in Romania [29]. In a Hungarian survey, participants who intended to reduce their meat intake in the next year had an expected increase of 1.47 in the number of preferred insect types that they would eat as a substitute for meat in the next year [40]. Similarly, a Belgian survey found that participants who intended to reduce their meat intake in the next year were 4.5 times more likely to be ready to adopt insects as a substitute for meat in the next year [44]. In the multi-national survey conducted in Germany, France, and the UK, participants who scored higher on the scale assessing commitment to meat provided lower ratings for the expected tastiness, healthiness, and environmental friendliness of pea and algae burgers [36]. Moreover, a 2021 Germany survey found that the higher a participant scored on the questionnaire evaluating attachment to meat, the less likely they were to choose the meat-hybrid option in which a portion of meat was replaced with a plant-based protein [33]. Similarly, in the DCE conducted in Germany and Belgium, the more attached a participant was to meat, the more likely they were to choose the 100% meat option compared to meat-hybrid options [34].

##### Consumption of Meat Substitutes

One study examined the effects of prior experience with meat substitutes and repeated exposure to meat substitutes on the long-term acceptance of non-meat protein sources as replacements for meat [42]. At the start of this longitudinal experiment in the Netherlands, participants liked Quorn^®^ and tofu less than the reference meat of chicken [42]. Although in general, the liking of Quorn^®^, tofu, and chicken decreased over the ten-week repeated exposure period, there was no difference in the decrease in liking of Quorn^®^, tofu, and chicken. Furthermore, the liking scores of Quorn^®^, tofu, and chicken were notably no longer different from one another after this ten-week repeated exposure period. Additionally, the number of boredom patterns, defined as a decrease in the liking of a product over time, and mere exposure patterns, defined as an increase in the liking of a product over time, differed among the three food products in this experiment. In contrast to the majority of participants who ate chicken and showed a boredom pattern over the repeated exposure period, the majority of participants who ate tofu showed a mere exposure pattern. On the other hand, the participants who ate Quorn^®^ took an intermediate position between chicken and tofu in terms of boredom and mere exposure patterns with the slight majority showing boredom over the repeated exposure period. Prior experience with meat substitutes was also associated with increased product liking of Quorn^®^ and tofu [42].

##### Cooking Habits and Food Involvement

Two studies explored cooking habits and food involvement as factors influencing the replacement of meat with non-meat protein sources [13,32]. In a recent US survey, cooking habits (i.e., the time and pleasure derived from cooking) were not found to be associated with a more positive assessment of blending mushrooms into traditional meat-based dishes [13]. However, food involvement (i.e., time spent thinking about food and time spent cooking and cleaning up after meals), was found to be associated with a more positive assessment of blending, which was associated with a greater acceptance of blending mushrooms into traditional meat-based dishes [13]. In a 2021 Germany survey, omnivores perceived meat as being easier to prepare than meat alternatives [32].

### 3.3. Socio-Cultural Factors

#### 3.3.1. Culture

##### Country of Consumer

Three studies examined the possible role of culture in the replacement of meat with non-meat protein sources by incorporating participants from different countries in their study designs [29,34,36]. In an online survey conducted in Denmark, Finland, Germany, Iceland, and Romania among female participants, the attitude towards using plant protein in food production was shown to influence both the intention to substitute meat in the diet as well as the attitude towards using rapeseed protein as an ingredient in meat analogues in all five countries [29]. Additionally, the attitude towards gluten was associated with a decreased intention to buy meat analogues in all five countries. For Germany only, however, the attitude towards soy protein was associated with an increased intention to buy meat analogues, whereas the attitude towards potato starch was associated with a decreased intention to buy meat analogues. In Romania, those eating less meat were more likely to buy meat analogues [29]. In the multi-national survey conducted in Germany, France, and the UK, it was found that pea and algae burgers were expected to be less tasty but healthier and more environmentally friendly than the beef burger [36]. Nevertheless, being from France was associated with providing lower ratings for the expected tastiness and healthiness of pea burgers compared to being from Germany. Contrastingly, the country of origin was not found to be associated with the ratings for expected tastiness, healthiness, and environmental friendliness of the algae burgers [36]. In a DCE conducted in Germany and Belgium, the majority of participants considered meat to be tastier than meat-hybrids yet considered meat-hybrids to be more environmentally friendly and better for animal welfare compared to meat [34]. Nevertheless, the majority of German participants considered meat-hybrids to be healthier than meat, whereas the majority of Belgian participants considered meat to be healthier than meat-hybrids [34].

#### 3.3.2. Social Norms, Roles, and Relationships

##### Situational Context

Two studies investigated the situational context of consumption as a factor influencing the replacement of meat with non-meat protein sources [30,32]. In a 2021 experiment in the Netherlands, photographs of meat and plant-based protein products were presented to 309 participants with a question regarding the appropriateness of the meat and plant-based meat substitutes during the hot meal of the day [30]. Overall, the plant-based meat substitutes and chickpeas and nuts were considered less appropriate than the corresponding meat product in almost all situations including: eating alone; with family and friends; cooking for children; to add flavor to a dish; or when there is little time for cooking. Besides the situation of cooking for a vegetarian in which the plant-based meat substitutes and chickpeas and nuts were rated as more appropriate than the corresponding meat product, the vegetarian hamburger was also rated as more appropriate than the hamburger when wanting to eat a healthy meal. Moreover, in situations of wanting to prepare a special meal, the steak was rated highly; but neither the hamburger nor the smoked sausage was rated higher than the corresponding vegetarian burger and vegetarian sausage in this situation [30]. In a 2021 Germany survey, omnivores and flexitarians alike rated the situational appropriateness of consuming a plant-based meat alternative to be most appropriate when eating alone [32]. For omnivores, eating plant-based meat alternatives in more formal settings (i.e., eating a Sunday dinner with family, invited for dinner at a restaurant, at a business meal, or when at a barbecue) was considered as less appropriate than in more casual settings (i.e., eating alone, being invited to eat with friends, or eating dinner with family during the weekday). For flexitarians, eating plant-based meat alternatives when eating alone or when eating with family during the weekday was considered more appropriate than more social or formal settings including: when invited to eat with friends; eating a Sunday dinner with family; invited for a dinner at a restaurant; at a business meal; or when at a barbecue [32].

### 3.4. External Factors

#### 3.4.1. Economic Factors

##### Price

Five studies assessed the role of price in the replacement of meat with non-meat protein sources [13,32,34,37,38]. In the 2016 UK DCE, the largest consumer segment was identified as price-conscious consumers making up 43% of meat-eaters and meat-reducers in the study [37]. For price-conscious consumers, price was the third most influential product attribute when determining their preference for ground meat or ground meat substitute preceded only by the type of ground meat or ground meat substitute and the region of production [37]. In the 2019 UK DCE, price was also found to be an influential factor in the preference of ground meat or ground meat substitute but only for some consumer segments of meat-eaters [38]. Among meat-eaters, price-conscious consumers were again the largest consumer segment making up 63% of all meat-eaters in the study. Price-conscious meat-eaters were influenced more by the type and price of ground meat or ground meat substitute, whereas price played a less influential role for traditional and empowered meat-eaters in the study. Among meat-reducers, price was found to be an intermediate factor for the health curtailers and the least influential factor for sustainable meat-reducers in determining their preference for ground meat or ground meat substitute [38]. In a 2021 Germany survey, however, both omnivores and flexitarians alike rated meat as performing better in terms of price compared to meat alternatives [32]. Furthermore, participants perceived steak as being the most expensive among tofu, chicken nuggets, vegetarian nuggets, wiener sausages, and vegetarian sausages [32].

In terms of partially replacing meat-products with plant-based protein, a recent US survey found that price benefits (i.e., reducing the cost of meals and helping with the budget) were rated as one of the top two reasons to consume blended meat products preceded only by health benefits [13]. In the DCE conducted in Germany and Belgium, a decrease in price was found to increase the preferability of the 100% meat and meat-hybrid options except for salami among Belgian participants [34].

#### 3.4.2. Food Environment

##### Extrinsic Product Attributes

Six studies investigated extrinsic product attributes (i.e., packaging, brand, nutritional information, health claims, and local origin and environmental labels), as factors in the replacement of meat with non-meat protein sources [13,31,34,37,38,44]. In a French experiment, participants preferred to purchase the pork-based sausage rather than the plant-based sausage after tasting and being provided with the packing information for the sausages, which included information on the brand, ingredients, nutrition, preparation, and recycling [31]. Nevertheless, although participants still preferred to purchase the pork-based sausage over the plant-based sausage, participants’ preference to purchase the plant-based sausage was higher than it had been during the blind tasting alone without the packaging information provided [31].

When considering the specific information provided on packaging, the 2016 UK DCE found that the brand and type of the ground meat or ground meat substitute primarily influenced taste-driven consumers, whereas brand was found to have only an intermediate or low influence on the other four consumer segments of meat-eaters and meat-reducers [37]. The 2019 UK DCE also found that brand had a low influence on the preference of ground meat or ground meat substitute in meat-eaters and meat-reducers [38]. In terms of health labels, the DCE in Germany and Belgium found that health labels had no effect on the preferability of the meat-hybrid options in which a portion of meat had been replaced by plant-based protein [34]. In the 2016 UK DCE, the fat content of ground meat and ground meat substitutes primarily influenced healthy and organic consumers but only intermediately influenced price-conscious, green, and taste-driven consumers [37]. In the 2019 UK DCE, the fat content of ground meat and ground meat substitutes intermediately influenced consumer segments of meat-eaters but more strongly influenced the health curtailers and sustainable consumers of meat-reducers [38]. For organic and local origin labels, the DCE in Germany and Belgium found that organic and local origin labels had a primarily positive effect on the preferability of the meat-hybrid options [34]. In the 2016 UK DCE, an origin label of the ground meat or ground meat substitute had a strong influence on the preferability of the ground meat or ground meat substitute for the price-conscious, green, taste-driven, and healthy consumers but was the least influential factor for organic consumers [37]. In the 2019 UK DCE, the origin label had an intermediate influence on the preferability of ground meat and ground meat substitute of meat-eaters and meat-reducers [38]. In terms of production labels, both the 2016 and 2019 UK DCEs found that production labels have an intermediate to weak influence on the preferability of ground meat and ground meat substitutes for meat-eaters and meat-reducers [37,38]. For environmental labels, the 2016 and 2019 UK DCEs also found that carbon footprint had an intermediate to low influence on the preferability of ground meat and ground meat substitute for most consumers besides green and sustainable consumers in which it was the primary influence [37,38]. In the DCE in Germany and Belgium, the environmental label had a positive effect on the preference of the meat-hybrid, except for the meat-hybrid options of meatballs and salami in Belgium [34].

Specifically, two studies specifically investigated the impact of the format of the replacement for meat on consumers’ preferences [13,44]. In a recent US survey, participants rated burgers as the most preferred format for consuming blended products in which traditional meat-based dishes are partially replaced by mushrooms followed by stir-fry with ground beef, meatloaf, tacos, chili with ground beef, and pasta with ground beef [13]. In a Belgian survey, it was found that every one unit increase in a participant’s orientation towards convenience in meal preparation was associated with a 75% increase in the readiness to adopt insects as a substitute for meat [44].

##### Meal Context

Four studies examined the possible role of meal context on the replacement of meat with non-meat protein sources [28,35,42,48]. In two experiments in the Netherlands, the liking of meat substitutes differed when evaluated individually or within the meal context [28,42]. In a 2011 experiment, participants liked Quorn^®^ pieces more than Quorn^®^ ground when evaluating these meat substitutes individually [28]. Participants also liked Quorn^®^ pieces more than Quorn^®^ ground in the rice and salad dishes; however, there were no differences in the participants’ liking of Quorn^®^ pieces and Quorn^®^ ground in the spaghetti and soup dishes. Despite these findings, there were no differences in participants’ overall liking of the rice, salad, spaghetti, or soup dishes using either Quorn^®^ pieces or Quorn^®^ mince [28]. Similarly, a longitudinal experiment found that participants liked the entire meals consisting of either Quorn^®^, tofu, or chicken better than Quorn^®^, tofu, or reference chicken evaluated individually outside of the meal context [42].

In two experiments conducted in the US, participants’ liking of dishes in which meat had been partially replaced by legumes differed depending upon the recipe and thus the meal context [35,48]. A 2018 US experiment found that the high-meat pork carnitas arepas were liked better in terms of overall liking compared to the low-meat arepas and the high- and low-meat chicken tikka masala in which a portion of the meat had been replaced by legumes in the low-meat recipes [48]. Nevertheless, there was no difference in the overall liking among the low-meat arepas and high- and low-meat chicken tikka masala recipes [48]. In a 2021 US experiment, there were no differences in the overall liking of high- and low-meat versions of East Asian bowls with regular and spicy sauces in which a portion of the beef was replaced with legumes and vegetables in the low-meat dishes [35].

##### Grocery Store Infrastructure

One study investigated the infrastructure of grocery stores as a factor influencing the replacement of meat with non-meat protein sources [46]. In a 2018 Canadian study, participants were interviewed on how well grocery stores in Canada support the transition to plant-based protein [46]. In terms of product availability, it was found that significantly more space was allocated to animal-based protein compared to plant-based protein in grocery stores. Some participants of the study elaborated that meat and dairy sections carry more types of products and brands in comparison to the lack of variety available for plant-based protein products, particularly tofu, grains, and legumes. For product promotion, there were more promotions for animal-based proteins than plant-based proteins in grocery stores. Furthermore, there was a higher percentage of sales and/or descriptive designated for animal-based proteins (32%) compared to plant-based proteins (3%), which was supported by participants noting that meat, seafood, and dairy sections were more “prominent” in grocery stores in contrast to plant-based proteins “hidden” locations throughout the grocery stores’ aisles [46]. In regard to product location, participants rated it easier to find animal-based protein sources than plant-based protein sources with one of the most commonly cited obstacles being the inconsistent location of plant-based proteins among different grocery stores [46].

#### 3.4.3. Animal Welfare

One study examined the role of animal welfare in the replacement of meat with non-meat protein sources [33]. In a 2021 German survey, the higher a participant rated meat-hybrids in terms of their animal welfare, the more likely they were to choose the meat-hybrid option consisting of 40% plant-based protein compared to the 100% meat option [33].

## 4. Discussion

Altogether, this review revealed multiple personal, socio-cultural, and external factors relating to the replacement of meat with non-meat protein sources among omnivores and flexitarians in developed countries. Most importantly, the results indicate that female gender, information on health and the environment, and lower price of non-meat protein sources may act as drivers to replacing meat with non-meat protein sources. Contrastingly, the results show that male gender, food neophobia, attachment to meat, and the perceived lower situational appropriateness of consuming non-meat protein sources in social settings may be inhibitors to replacing meat with non-meat protein sources. Interestingly, although sensory and hedonic attributes of meat such as taste and flavor may act as inhibitors, the recipe and entire meal context appear to be more important than the individual evaluation of non-meat protein sources and thus may act as a driver increasing consumers’ acceptability of non-meat protein sources in traditionally meat-based dishes.

Notably, gender, food neophobia, and information on health and the environment are among the factors most researched in literature for their role in reducing meat intake and consuming non-meat protein sources [16,49,50,51,52,53,54,55]. Similar to the findings in this review, many studies have shown that gender influences attitudes related to the consumption of meat and non-meat protein sources [16,49,50,51]. Considering common phrases such as “real men eat meat”, studies have shown that men who identified more with traditional beliefs of masculinity that conflate meat consumption and virility were more attached to meat and had a more negative attitude towards a vegetarian diet [50,51]. In accordance with our findings, a recent systematic review incorporating ninety-one articles on consumer acceptance of novel alternative protein sources also highlighted that food neophobia remains a hindrance for consumers to adopt many novel protein sources including insects, seaweed, cultured meat, and plant-based meat substitutes into daily consumption patterns [52]. In terms of information on health and the environment, studies have shown that providing consumers with information on the negative health and environmental consequences of meat consumption increased intentions to reduce meat consumption [53,54,55], which aligns with our findings.

While this review identified many factors relating to reducing meat consumption, some potentially important factors cannot be substantiated due to mixed results, not being extensively examined, or being entirely missing from the included studies. Age and socioeconomic status have both been cited as influencers of meat consumption in literature [56,57]; yet, findings on these factors were mixed in this review and could not confirm younger age and higher socioeconomic status as being drivers for replacing meat with non-meat protein sources. Going further, only one study each examined sex [36], grocery-store infrastructure [46], and cooking skills [14], making it impossible to definitively draw conclusions on whether these factors act as drivers or inhibitors to replacing meat with non-meat protein sources. Moreover, race, ethnicity, religion, health status, degree of urbanization, and perceived behavior control were not examined by any of the studies included in this review. Since previous studies have shown differences in the consumption of meat and specific meat products among white, Hispanic, and African Americans [58,59], race and ethnicity could be suspected of also influencing the replacement of meat with non-meat protein sources. Additionally, examining perceived behavior control could be instrumental in determining how and to what degree certain knowledge and skills, such as cooking skills, augment consumers’ willingness and self-efficacy in being able to institute non-meat protein sources as dietary fixtures [60].

Besides examining the relevance of the aforementioned underresearched factors, studies should prioritize standardizing methods and examining potential drivers and inhibitors across different consumer segments and types of non-meat protein sources in order to foster comparability among studies as well as to identify variations in consumer acceptance and long-term health and environmental consequences. Importantly, studies should standardize their definitions of “meat” namely whether this excludes certain types of meat and includes poultry, fish, and seafood. For replacements of meat, terms such as “plant-based diet”, “vegetarian meals”, or “meat-less meals” lack adequate description needed for reproducibility in studies and furthermore do not necessitate having sufficient protein content to constitute an actual replacement of meat within a meal context. In regard to study populations, utilizing random versus convenience sampling would provide a more accurate depiction of the population and would avoid volunteer bias [61]. Separate analyses of omnivores, flexitarians, pescatarians, vegetarians, and vegans would also be advantageous in determining subtle distinctions in motivations, willingness, and acceptance that could be employed to build more efficacious public health campaigns to reduce meat [62]. Although most studies included men and women in their analyses, future studies should be explicit in whether they are examining biological differences between the male and female sexes or the psychosocial and cultural differences of male, female, and other genders as implications of such research differ greatly [63]. Beyond different consumer segments, studies should also incorporate various types of non-meat protein sources of different processing levels in order to determine differences in consumer acceptance as well as to forecast the long-term health and environmental consequences of such replacements [64,65]. When possible, it is essential that we amass more experimental evidence in real-life settings to assess if and how these factors truly affect the motivation, willingness, and acceptance of replacing meat with non-meat protein sources.

Ultimately, this review has important implications ranging from public health policy to research collaboration. Firstly, the findings from this review identify relevant drivers and inhibitors that can be used to support more efficacious public health campaigns aiming to reduce meat consumption within developed countries. Next, this research may be relevant for food industries when marketing non-meat protein sources to consumers as replacements for meat during mealtimes. For dietitians and other healthcare professionals, this research could be used as a tool to assist clients and patients in fostering behavior change towards healthier and more sustainable food options. Besides the aforementioned considerations for future research, the findings in this review on the relevance of meal context emphasize the importance of collaboration within the field of nutrition particularly among chefs, dietitians, and scientists to create flavorful dishes with non-meat protein sources in an effort to facilitate and expedite the transition to healthier and more sustainable protein sources among consumers.

Nevertheless, this review has some limitations that should be noted. Although beyond the scope of this review, replacing red and processed meat with white meat, eggs, or dairy products may be a more feasible first step for many consumers attempting to reduce meat consumption [66]. Additionally, this review focused on omnivores and flexitarians given that they comprise a much larger percentage of the population in developed countries compared to pescatarians, vegetarians, and vegans [22,23,26]. However, understanding the motivations, willingness, and behavior of these non-meat-eating subgroups and comparing them with omnivores and flexitarians could be useful in identifying which factors are most decisive in reducing meat. In this review, one researcher screened articles for inclusion and exclusion criteria and extracted relevant data. Although this could have introduced bias in the article section, the inclusion and exclusion criteria were clearly defined, and the search strings were built by two researchers. Furthermore, a second researcher randomly cross-checked article screening and data extraction and discussed any uncertainties or disagreements with the first researcher. Other frameworks, such as the Capability, Opportunity, Motivation behavior model (COM-B) could have been utilized to organize and summarize the findings in this review [67]. However, we chose to use the theoretical framework by Stoll-Kleemann and Schmidt (2017) as it provides a comprehensive overview of the interrelated drivers and inhibitors that may be involved specifically in meat-eating behavior, which relates closely to our aim of identifying the drivers and inhibitors of replacing meat with non-meat protein sources among consumers in developed countries [17]. Furthermore, unlike the COM-B model, this framework is based on a pro-environmental behavior model that analyzes the propensity of individuals to partake in actions that mitigate a negative impact on the environment as well as the dissonance between having environmental awareness and participating in pro-environmental behavior [27]. Yet, this framework does carry some shortcomings [17]. While not covered by the studies in this review or explicitly included in the framework [17], we included race and ethnicity as potentially important factors to consider in the socio-demographics in Table 2.

## 5. Conclusions

In conclusion, this review revealed multiple personal, socio-cultural, and external factors relating to the replacement of meat with non-meat proteins sources among omnivores and flexitarians in developed countries. The results indicate that female gender, information on health and the environment, and lower price of non-meat protein sources may act as drivers, whereas male gender, food neophobia, attachment to meat, and the lower situational appropriateness of consuming non-meat protein sources act as inhibitors. According to literature, gender, food neophobia, and information on health and the environment are relevant factors in reducing meat and replacing it with non-meat protein sources [16,49,50,51,52,53,54,55]. However, more research is needed to establish the relevance of socioeconomic status, race, ethnicity, religion, health status, food environment, and cooking skills. Future research should consider the importance of standardizing methods in order to allow for better comparisons among studies. Additionally, studies should prioritize examining potential drivers and inhibitors across different consumer segments and various non-meat protein sources to determine differences in consumer acceptability and the long-term health and environmental consequences of such replacements. Ultimately, the findings of this review are relevant for supporting more efficacious public health campaigns, product development and marketing in food industry, and behavior change facilitated by healthcare professionals. Given the importance of meal context, this research calls for collaboration particularly among chefs, dietitians, and scientists in research to expedite and facilitate the transition to healthier and more sustainable protein sources.

## Figures and Tables

**Figure 1 nutrients-13-03602-f001:**
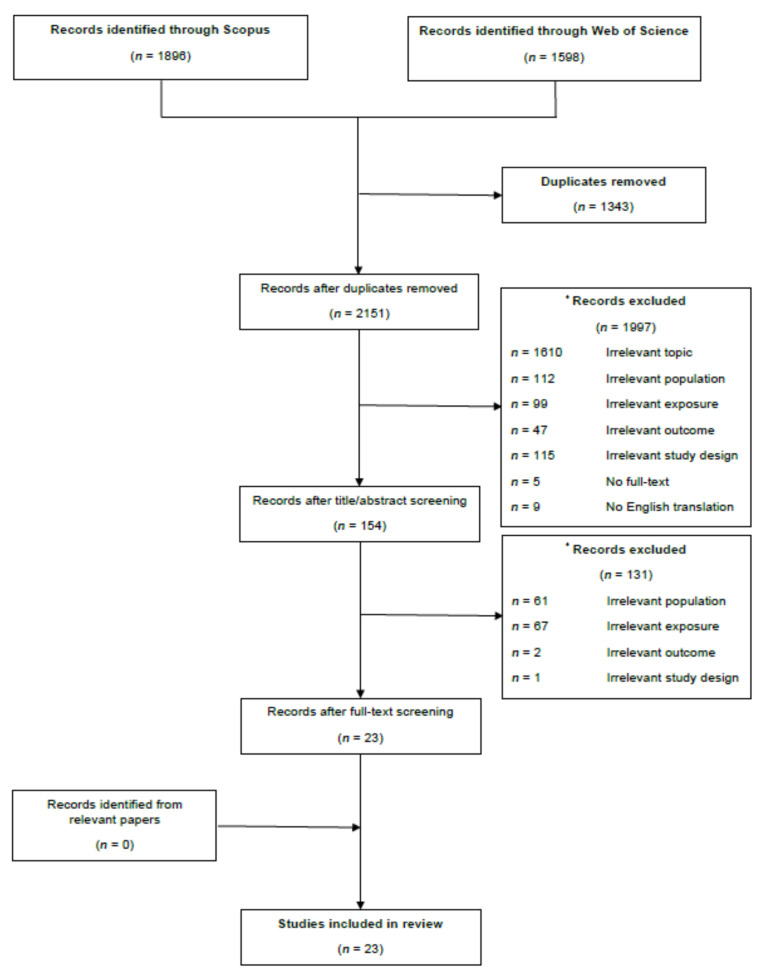
Flowchart of the identification, screening, and inclusion of studies assessing the drivers and inhibitors to replacing meat consumption with non-meat protein sources. * Exclusion criteria for review: (1) irrelevant topic (e.g., food production, animal physiology, or agriculture); (2) irrelevant population (e.g., included vegans, vegetarians, or pescatarians or only specialized populations such as students or armed forces); (3) irrelevant exposure (i.e., lack of replacement of meat with non-meat protein sources or nudging interventions); (4) irrelevant outcome (i.e., not pertaining to the perception, awareness, attitude, intent, willingness, or behavior to replace meat with non-meat protein sources); (5) irrelevant study design (i.e., reviews, protocols, pilot studies, editorials, opinions, or conference proceedings); (6) no full text available; and (7) no English translation available.

**Figure 2 nutrients-13-03602-f002:**
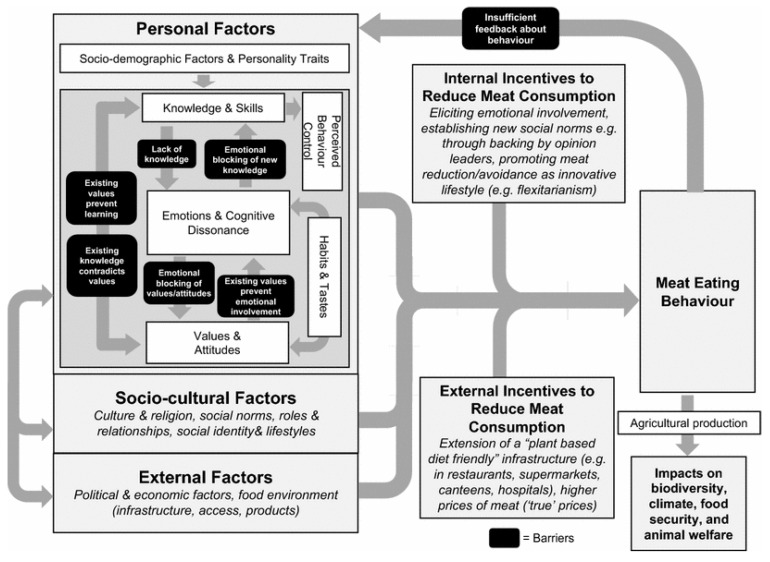
Model of factors that influence meat-eating behaviors. Reprinted from Stoll-Kleemann, S.; Schmidt, U.J. Reducing meat consumption in developed and transition countries to counter climate change and biodiversity loss: a review of influence factors. *Regional Environmental Change*
**2017,** *17* (5), 1261–1277. No changes were made to this figure. Creative Common License 4.0 International License available at https://creativecommons.org/licenses/by/4.0/. (accessed on 12 October 2021).

**Table 1 nutrients-13-03602-t001:** Overview of twenty-three studies included in the review examining the drivers and inhibitors underlying the replacement of meat with non-meat protein sources.

Author, Year	Study Design, Location, and Population	Research Aim	Non-Meat Protein Replacement(s)	Explanatory Variable(s)	Dependent Variables(s)	Main Outcome(s)
Apostolidis & McLeay (2016) [37]	DCE, UK; *n* = 233; men & women *	Identify the attributes of ground meat (substitute) that influence consumers’ choices	Ground meat substitute (i.e., soy, tofu, Quorn^®^, etc.)	Type of ground meat (substitute) Fat content of ground meat (substitute) Carbon footprint of ground meat (substitute) Method of production of ground meat (substitute) Price of ground meat (substitute) Origin of ground meat (substitute) Brand of ground meat (substitute) Age Gender Income Household Region of residence	Preference for attributes of ground meat (substitute)	Five consumer segments were identified among meat-eaters and meat-reducers: price-conscious (42.5%), green (17%), taste-driven (14.6%), healthy (10.5%), and organic (9.7%) consumers. Strongest influences for price-conscious consumers were type of ground meat (substitute) and origin; for green consumers, carbon footprint and origin; for taste-driven consumers, type of ground meat (substitute) and brand; for healthy consumers, fat content and type of ground meat (substitute); and for organic consumers, fat content and type of ground meat (substitute).
Apostolidis & McLeay (2019) [38]	DCE, UK; *n* = 400; 61% women *	Compare the importance of sustainability-related labels on consumers’ preferences for ground meat (substitute)	Ground meat substitute (Quorn^®^)	Type of ground meat (substitute) Fat content of ground meat (substitute) Carbon footprint of ground meat (substitute) Method of production of ground meat (substitute) Price of ground meat (substitute) Origin of ground meat (substitute) Brand of ground meat (substitute) Age Gender Income Household Region of residence	Preference for sustainability labels of ground meat (substitute)	Meat-eaters were primarily influenced by the type of ground meat (substitute), origin, and price, whereas the meat-reducers were primarily influenced by the type of ground meat (substitute) fat content, and origin. Meat-eaters were segmented into three consumer segments: price-conscious (63%) primarily influenced by the type of ground meat (substitute) and price; traditional (19%) primarily influenced by the type of ground meat (substitute) and origin; and empowered (18%) primarily influenced by type of ground meat (substitute) and production. Meat-reducers were segmented into two consumer segments: health curtailers (82%) primarily influenced by the fat and origin; and sustainable (18%) primarily influenced by carbon footprint and fat content.
Banovic & Sviensdóttir (2021) [29]	Online survey, Denmark, Finland, Germany, Iceland, and Romania; *n* = 1397; 100% women *	Investigate whether general attitudes towards using plant protein in food production and intention to substitute meat protein in the diet are related to the attitude towards rapeseed protein and the attitude and intention to buy meat analogues	Wiener sausages containing rapeseed protein	Attitude towards plant protein in food production Attitude towards rapeseed protein Attitude towards soy protein Attitude towards potato starch Attitude towards gluten Intention to substitute meat protein in the diet Meat consumption frequency Country of origin	Attitude towards and intention to buy meat analogues	Attitude towards using plant protein in food production was shown to influence both the intention to substitute meat and the attitude towards using rapeseed protein as an ingredient in meat analogues in all countries. Attitude towards rapeseed was shown to influence the attitude towards meat analogue.
Castellari et al. (2019) [39]	WTP experiment, Italy; *n* = 119; 47% women *	Evaluate the impact of explanatory messages about health and environment on consumers’ WTP for a beef burger and soy burger	Soy burger	Information on health Information on environment	WTP and chosen quantities for beef and soy burgers	Successive rounds of explanatory messaging on health and the environment resulted in a relative decrease of −1.6% in the WTP for beef burgers and relative increase of +3.6% in the WTP soy burgers. Successive rounds of explanatory messaging on health and the environment resulted in a relative decrease of −23.0% in the chosen quantities of the beef burger and a relative increase of +45.6% in the chosen quantities of soy burgers.
Elzerman et al. (2011) [28]	Experiment, The Netherlands; *n* = 93; 77% women ^§^	Obtain insight into the influence of meal context on the acceptance of meat substitutes	Quorn^®^ pieces Quorn^®^ ground Tofu strips Tivall^®^ stir-fry pieces Goodbite^®^ chicken style Vivera^®^ vega stir-fry pieces	Meal context of meat substitute Flavor of meat substitute Texture of meat substitute Form of meat substitute	Overall liking of meat substitute Overall liking of dish Product liking of meat substitute in dish Perceived appropriateness of meat substitute in meal context Intention to use dish with meat substitute	Quorn^®^ pieces were liked more than Quorn^®^ ground when compared separately in the rice and salad dishes, but there was no difference in the overall liking of the meals using Quorn^®^ pieces or Quorn^®^ ground. Shape of meat substitutes appears to influence the appropriateness of the meal more than the flavor and texture of the specific meat substitute.
Elzerman et al. (2021) [30]	Experiment, The Netherlands; *n* = 309; men & women *	Explore the perceived appropriateness of meat products, meat substitutes, and meat alternatives in different usage situations	Vegetarian ground Vegetarian stir-fry pieces Vegetarian hamburger Vegetarian sausages Chickpeas and nuts	Family context Special meal context Vegetarian context Friends context Alone context Children context Flavor context Little time context Health context	Perceived situational appropriateness of meat substitutes in various situations	Situational appropriateness of meat products was higher than meat substitutes and chickpeas and nuts in almost all situations except for vegetarian and health contexts.
Gere et al. (2017) [40]	Online survey, Hungary; *n* = 400; men & women *	Assess the readiness of Hungarian consumers to adopt insects as a substitute for meat	Insects	Age Gender Education Food neophobia Food technology neophobia Attitude towards health characteristics of food Convenience orientation for food choice Attention to the environmental impact of food Belief that meat is nutritious and healthy Intention to reduce fresh meat intake Familiarities with insects Familiarity with whey Familiarity with algae Familiarity with soy	Readiness to adopt insects as a substitute for meat	Participants intending to reduce their fresh meat intake within the next year had an expected increase of 1.47 in the number of preferred insect types they would be willing to consume as a substitute for meat. Food neophobia was found to be a barrier to the readiness to adopt insects as a substitute for meat.
Gravely & Fraser (2018) [46]	Interviews, Canada; *n* = 24	Investigate the in-store context for purchasing plant-based protein in major Canadian supermarkets	Plant-based protein	Product availability Product promotions Product location	Extent to which grocery stores are mediating the transition to plant-based protein sources	More space and promotions were allotted to animal-based protein than plant-based protein in grocery stores. Participants found it easier to find animal-based protein compared to plant-based protein in grocery stores.
Guinard et al. (2016) [47]	Experiment, USA; *n* = 147; 58% women *	Test consumer acceptance of meat-based dishes in which meat had been substituted with mushrooms	Low-meat carne asada ^†^ Low-meat beef tacos ^†^	Percent substitution with mushrooms Appearance of dish Flavor of dish Texture of dish	Overall liking Liking of appearance Liking of flavor Liking of texture Level of saltiness Level of spiciness Level of moistness	100% beef carne asada was liked more for overall liking, appearance, flavor, and texture compared to the 50% beef carne asada. 100% beef tacos were liked more than mushroom-containing beef tacos for appearance but not more for overall liking, flavor, and texture.
Hartmann & Siegrist (2020) [41]	Online survey, Germany; *n* = 973; 49% women *	Investigate the impact of unapologetic and apologetic justification strategies on consumers’ willingness to substitute meat with meat alternatives	Quorn^®^ Tofu Seitan Soy schnitzel	Unapologetic justification strategy: Pro-meat Denial Hierarchical justification Religion justification Health justification Human destiny Slaughter justification Apologetic justification strategy: Dissociation Dichotomization Avoidance	Willingness to substitute meat for Quorn^®^, tofu, seitan, or soy schnitzel	Participants who scored higher on the unapologetic justification strategies were less willing to substitute meat with Quorn^®^, tofu, seitan, or soy schnitzel compared to those who scored lower.
Hoek et al. (2013) [42]	Longitudinal experiment, The Netherlands; *n* = 89; 78% women ^¶^	Investigate the hedonic effects of repeated exposure to meat substitutes and meat	Quorn^®^ Tofu	Meal context Type of product Prior consumption of meat substitutes Prior consumption of chicken Repeated exposure to meat or meat substitute Different meals used Hunger Food neophobia Variety-seeking	Desire to eat the product Liking of the product Boredom with the product Amount of eaten product	Liking scores among Quorn^®^, tofu, and chicken were not different after the repeated exposure period. Most participants who ate tofu showed a mere exposure pattern of increased liking over time, whereas most participants who ate chicken showed a boredom pattern of decreased liking over time. Entire meal was liked better than Quorn^®^, tofu, or chicken evaluated separately. Food neophobia and variety-seeking did not have an effect on overall product liking over time.
Kemper & White (2021) [14]	Semi-structured interviews, New Zealand; *n* = 23; 74% women ^¶^	Explore young adults’ motivations, strategies, and barriers towards flexitarianism	Legumes Lentils Tofu	Cooking skills	Ability to adopt a flexitarian lifestyle	Participants who were more confident and experienced in cooking substituted meat for legumes, lentils, and tofu, whereas participants who were less confident and experienced in cooking preferred substituting meat with meat substitutes like vegetarian patties.
Lang (2020) [13]	Online survey, USA; *n* = 602 *	Explore consumers’ response to blending mushrooms into traditional meat-based foods and their lifestyle and motivations influencing the assessment and acceptance of these blended foods	Meat-hybrid products **	Perceived health benefits Perceived cost benefits Perceived taste benefits Perceived culinary benefits Perceived sustainability benefits Assessment of blending Format of blended products Red meat consumption Healthy eating Cooking habits Food innovativeness Food involvement Food knowledge Age Gender Income Education	Acceptance of blending mushrooms into traditionally meat-based foods	Top reasons for consuming blended foods were health benefits followed by price, taste, culinary, and sustainability benefits. Burgers were the preferred format for consuming blended products followed by stir-fry with ground beef, meatloaf, tacos, chili with ground beef, pasta with ground beef, and other. Age, gender, income, and education were not associated with the acceptance of the blending concept, but women assessed blending more positively than men. Participants whose red meat consumption was declining or were contemplating decreasing their consumption were associated with more favorable assessment and acceptance of blending.
Mancini & Antonioli (2019) [43]	Online survey, Italy; *n* = 485; men & women *	Assess the extent to which Italian consumers are willing to accept cultured meat	Cultured meat	Information on positive internalities of cultured meat Information on positive externalities of cultured meat	Perception of cultured meat	Participants showed better agreement with the extrinsic attributes of cultured meat compared to the intrinsic attributes.
Martin et al. (2021) [31]	Experiment, France; *n* = 102; 51% women ^¶^	Test if information concerning the consequences on health or the environment could be useful in promoting plant-based products	Plant-based sausage	Taste Packaging Information on health Information on environment	Preference to purchase plant-based sausage WTP plant-based sausage	Participants preferred to purchase the pork-based sausage over the plant-based sausage after the blind tasting and tasting with packaging. WTP for the plant-based sausage increased after the second message on health or the environment, and WTP for the pork-based sausage decreased after the second message on the environment.
Michel, Hartmann, et al. (2021) [32]	Online survey, Germany; *n* = 967; 50% women ^¶^	Identify barriers that prevent consumers from eating meat alternatives	Meat alternatives Vegetarian nuggets Tofu Vegetarian sausage	Eating alone context Eating with friends context Eating with family on a weekday context Eating with family on Sunday context Invited for dinner in a restaurant context Business meal context Barbecue party context Perceived taste Perceived texture Perceived price Perceived ease of preparation Perceived protein content Perceived fat content Perceived environmental friendliness Perceived masculinity Perceived festivity Perceived healthiness Perceived satiation Perceived naturalness Type of product	Acceptability of eating plant-based meat alternatives	Omnivores and flexitarians rated eating alone as the most appropriate situation to consume meat alternatives. Omnivores rated meat as performing better in regards to taste, texture, price, ease of preparation, protein content, fat content, and environmental friendliness, whereas flexitarians rated meat alternatives as performing better in terms of fat content and environmental friendliness. Participants perceived steak as being the most healthy, protein-rich, filling, natural, festive, masculine, and tasty.
Michel, Knaapila, et al. (2021) [36]	Online survey, Germany, France, and UK; *n* = 1734; 48% women ^¶^	Investigate the taste, healthiness, and environmental expectations of pea and algae burgers as meat alternatives and the factors influencing these expectations	Pea burger Algae burger	Age Sex Country of origin Meat commitment Food neophobia Attitude towards vegans and vegetarians	Expected tastiness Expected healthiness Expected environmental friendliness	Pea and algae burgers were expected to be healthier and more environmentally friendly but less tasty than beef burgers in all countries. Participants who were more committed to meat, food neophobic, and had a negative attitude towards vegans and vegetarians rated the tastiness, healthiness, and environmental friendliness of pea and algae burgers lower. Being older, male, and from France was associated with providing negative ratings for the tastiness, healthiness, and environmental friendliness of pea burgers.
Profeta, Baune, Smetana, Bornkessel, et al. (2021) [33]	Online survey, Germany; *n* = 500; 51% women *	Identify consumer attitudes and preferences for meat-hybrids	Meat-hybrid products ^††^	Perceived tastiness Perceived healthiness Perceived environmental friendliness Perceived animal welfare Attachment to meat Food neophobia Frequency of purchasing plant-based alternatives	Preference for meat-hybrids	Participants rated meat-hybrids as performing better in terms of perceived healthiness, environmental friendliness, and animal welfare but performing worse in terms of perceived tastiness compared to the 100% meat option. The more attached a participant was to meat and the more food neophobic, the less likely they were to choose the meat-hybrid. The higher the participant rated the meat-hybrid in terms of perceived healthiness, environmental friendliness, and animal welfare, the more likely they were to choose the meat-hybrid with perceived healthiness exerting the largest influence.
Profeta, Baune, Smetana, Broucke, et al. (2021) [34]	DCE, Germany and Belgium; *n* = 1001; 51% women *	Identify consumer attitudes and preferences for meat-hybrids	Meat-hybrid meatballs ^††^ Meat-hybrid mortadella ^††^ Meat-hybrid salami ^††^ Meat-hybrid chicken nuggets ^††^ Vegetarian meatballs Vegetarian mortadella Vegetarian salami Vegetarian chicken nuggets	Perceived tastiness Perceived healthiness Perceived environmental friendliness Perceived animal welfare Percent substitution with plant-based protein Organic label Origin label Environmental label Nutritional label Price Attachment to meat Food neophobia Importance of eating healthy	Preference for meat-hybrids	Participants in Germany rated the meat-hybrids as performing better in terms of perceived healthiness, environmental friendliness, and animal welfare, whereas participants in Belgium rated meat-hybrids as performing better in terms of perceived environmental friendliness and animal welfare. Meat was the most preferred option followed by the meat-hybrids with the least preferred option being the 100% vegetarian products.
Spencer et al. (2018) [48]	Experiment, USA; *n* = 110; 58% women ^§^	Test the concept of the Flexitarian Flip^TM^ in a dining venue context by replacing meat with legumes in meat-based recipes	Low-meat pork carnitas arepas ^†^ Low-meat chicken tikka masala ^†^	Amount of meat in recipe Flavor Texture Appearance Spiciness	Overall liking of dish Liking of appearance of dish Liking of flavor of dish Liking of texture of dish Liking of spiciness of dish	High-meat arepas were liked more than low-meat arepas and high- and low-meat chicken tikka masala dishes for overall liking. High-meat dishes were liked more than low-meat dishes for overall liking and flavor liking. Spicy versions of the arepas and chicken tikka masala recipes were liked more for flavor and texture than the regular versions across all meat levels.
Spencer et al. (2021) [35]	Experiment, USA; *n* = 144; 65% women; * 58% Caucasian	Investigate implementation of the mixed dish Flexitarian Flip^TM^ strategy in a different geographical area and with a new cuisine	Low-meat East Asian bowls ^†^	Amount of meat in recipe Flavor Texture Appearance Spiciness Satiation Satisfaction Gender	Overall liking of recipe Liking of appearance of recipe Liking of flavor of recipe Liking of texture of recipe Liking of spiciness	No differences in the overall liking, flavor, texture, appearance, satiation, or satisfaction of bowls regardless of the amount of meat. Across all subjects and bowls, not having enough flavor complexity resulted in a decrease in overall liking.
Verbeke (2015) [44]	Online survey, Belgium; *n* = 368; 61% women *	Profile consumers who claim to be ready to eat insects as a substitute for meat	Insects	Age Gender Education Familiarity with the idea of eating insects Food neophobia Food technology neophobia Attitude towards health characteristics of food Convenience orientation for food choice Attention to the environmental impact of food Belief that meat is nutritious and healthy Importance of taste when evaluating meat Intention to reduce fresh meat	Readiness to adopt insects as a substitute for meat	Food neophobia was the largest barrier to being ready to adopt insects as a substitute for meat. Male gender, familiarity with the idea of eating insects, convenience orientation for food choice, attention to the environmental impact of food, and planning on reducing fresh meat intake within the next year all increased the likelihood of readiness to adopt insects as a substitute for meat. Increase in age, food neophobia, food technology neophobia, belief that meat is nutritious and healthy, and the importance of taste when evaluating meat all decreased the likelihood of readiness to adopt insects as a substitute for meat.
Weinrich et al. (2020) [45]	Online survey, Germany; *n* = 713; 53% women *	Explore the readiness and intentions of consumers to use cultured meat, their attitudes and their driver strength, and demographic predictors	Cultured meat	Attitude towards cultured meat Age Gender Education Income Region Pre-knowledge Living with children	Intention to try and eat cultured meat and intention to promote cultured meat to friends	Participants’ attitudes towards cultured meat were structured into three dimensions: ethics, emotional objections, and global diffusion optimism. Ethics was the strongest predictor for using cultured meat followed by emotional objections and global diffusion. Participants’ pre-knowledge of cultured meat was shown to increase the ethical beliefs of cultured meat but did not impact the emotional objections or the global diffusion optimism of cultured meat.

* Study specified investigating gender. ^†^ Traditionally meat-based recipes in which a portion of the meat had been substituted with legumes.^. §^ Study did not specify whether gender or sex was investigated. ^¶^ Study specified investigating sex. ** Meat-hybrids consisting of meat-based food products in which a partial portion of the meat had been replaced by mushrooms. ^††^ Meat-hybrids consisting of meat-based food products in which a partial portion of the meat had been replaced by plant-based protein. Abbreviations: DCE, discrete choice experiment; WTP, willingness to pay; UK, United Kingdom; USA, United States of America.

**Table 2 nutrients-13-03602-t002:** Summary of drivers and inhibitors underlying the replacement of meat with non-protein sources examined by the studies included in this review *.

Factors			Summary	Number of Studies Examining the Factor
Personal	Socio-demographics	Age [13,36,37,38,44,45]	Unclear whether age is a driver or inhibitor.	6
		Gender and sex [13,35,36,37,38,44,45]	Female gender may be a driver and male gender may be an inhibitor to replacing meat with non-meat protein sources, but it is unclear if this applies to all alternative protein sources such as insects and cultured meat. Male sex is a possible inhibitor to replacing meat with non-meat protein sources but was only examined by one study.	7
		Socioeconomic status [13,37,38,44,45]	Unclear whether socioeconomic status is a driver or inhibitor.	5
		Ethnicity and race	Factor was not examined by the studies in this review.	0
		Religion	Factor was not examined by the studies in this review.	0
	Sensory and hedonic aspects	Taste, texture, and appearance [13,28,31,32,35,44,47,48]	Taste and texture of meat may be inhibitors to replacing meat with non-meat protein sources depending upon the recipe and meal context, but appearance of a dish appears to exert much less of an influence.	8
	Hunger cues	Hunger and satiety [32,35,42]	Hunger and satiety may be drivers to replacing meat with non-meat protein sources.	3
	Personality traits	Food and food production neophobia [13,33,34,36,40,42,44]	Food neophobia may be an inhibitor to replacing meat with non-meat protein sources, particularly for novel alternative protein sources.	7
		Variety seeking [42]	Variety seeking may be an inhibitor to replacing meat with non-meat protein sources but was only examined by one study.	1
	Knowledge and skills	Information on health and the environment [31,39,43]	Providing information on health and the environment may be drivers to replacing meat with non-meat protein sources.	3
		Cooking skills and food knowledge [13,14,44,45]	Food knowledge of non-meat protein sources may be a driver to replacing meat with non-meat protein sources. Cooking skill is a possible driver to replacing meat with non-meat protein sources but was only examined by one study.	4
	Emotions and cognitive dissonance	Justification strategies to consume meat [41]	Unapologetic justification strategies are a possible inhibitor to replacing meat with non-meat protein sources but were only examined by one study.	1
	Values and attitudes	Health and the environment [13,32,33,34,44]	Health may be a driver or inhibitor to replacing meat with non-meat protein sources depending on whether meat or the non-meat protein source is perceived as being healthy by the consumer. Sustainability may be a driver to replacing meat with non-meat protein sources but to a lesser degree than health.	5
		Plant protein sources and food production [13,29,45]	More positive assessments of plant protein sources and production may be drivers to replacing meat with non-meat protein sources.	3
		Vegans and vegetarians [36]	Negative attitude towards vegans and vegetarians is a possible inhibitor to replacing meat with non-meat protein sources but was only examined by one study. The influence of a positive attitude towards vegans and vegetarians was not examined.	1
		Others [13,32]	Perceived naturalness, masculinity, and festivity are possible inhibitors to replacing meat with non-meat protein sources but were only examined by one study. Perceived culinary benefits do not appear to be a driver to replace meat with non-meat protein sources but was only examined by one study.	2
	Habits and tastes	Healthy eating [13]	Healthy eating is a possible driver to replacing meat with non-meat protein sources but was only examined by one study.	1
		Consumption of meat [29,33,34,36,40,44]	Consuming less meat may be a driver whereas increased commitment or attachment to meat may be an inhibitor to replacing meat with non-meat protein sources.	6
		Consumption of meat substitutes [42]	Consumption of meat substitutes is a possible driver to replacing meat with non-meat protein sources but was only examined by one study.	1
		Cooking habits and food involvement [13,32]	Unclear whether cooking habits or food involvement is a driver or inhibitor.	2
	Perceived behavior control		Factor was not examined by the studies in this review.	0
Socio-cultural	Culture	Country of consumer [29,34,36]	Specific country of participant may be a driver or inhibitor to replacing meat with non-meat protein sources.	3
	Social norms, roles, and relationships	Situational context [30,32]	Perceived lower situational appropriateness of non-meat protein sources may be an inhibitor to replacing meat with non-meat protein sources particularly in formal and social contexts.	2
	Social identity and relationships		Factor was not examined by the studies in this review.	0
External	Political factors		Factor was not examined by the studies in this review.	0
	Economic factors	Price [13,32,34,37,38]	Lower price of non-meat protein sources may be a driver to replacing meat with non-meat protein sources but possibly only for specific consumer segments.	5
	Food environment	External product-attributes [13,31,34,37,38,44]	Packaging information, format, and convenience packaging of the non-meat protein source may be drivers to replacing meat with non-meat protein sources but possibly only for specific consumer segments.	6
		Meal context [28,35,42,48]	Meal context may be a driver to replacing meat with non-meat protein sources and is likely a more pertinent factor in acceptability than evaluating the individual non-meat protein source alone.	4
		Grocery-store context [46]	Grocery store context is a possible inhibitor to replacing meat with non-meat protein sources but was only examined by one study.	1
	Animal welfare	Animal welfare [33]	Animal welfare is a possible driver to replacing meat with non-meat protein sources but was only examined by one study.	1

* Grouping of drivers and inhibitors based on the model of factors that influence meat-eating behaviors from Stoll-Kleemann, S.; Schmidt, U.J. Reducing meat consumption in developed and transition countries to counter climate change and biodiversity loss: a review of influence factors. *Regional Environmental Change*
**2017**, *17* (5), 1261–1277. Creative Common License 4.0 International License available at https://creativecommons.org/licenses/by/4.0/. (accessed on 12 October 2021).

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
