# Peer review of "Replacement of Meat with Non-Meat Protein Sources: A Review of the Drivers and Inhibitors in Developed Countries"

_nutrients, 2021, doi:10.3390/nu13103602_

Round 1
Reviewer 1 Report
Line 77 – should be flexitarian(s) (make plural)
Figure 1 – make text larger in boxes – improve graphic quality
Table 1 is to big and bulky. Suggest you put 2 most significant papers in the table as examples and move the rest to an appendix. Or include a much more limited version in the paper with the full table as appendix.
Overall, beautifully written.
Reviewer 2 Report
This is a mostly well-written review on an interesting and increasingly important topic – the current evidence and evidence gaps around the drivers and inhibitor of replacing meat with meat-substitutes.
Having said that, I have two main criticisms that I would suggest the authors address:
Major issues:
- Support and discussion of the theoretical model used. Although there is a brief mention later in the paper about an alternative model, other than the single paragraph lines 106-113, there is really no discussion at all in the introduction regarding existing models, their strengths and weaknesses, why the Stoll-Kleemann and Schmidt (2017) model was chosen, etc. This is really important, particularly for a reader not already familiar with this literature.
- My second major criticism, and related to the first, is that there is little context provided around the components of the model. Each is simply listed as a sub-heading in the Results section and then a count/description of papers that examine that factor is listed. For example, “cooking habits and food involvement”; “emotions and cognitive dissonance”; “extrinsic product attributes”; “food neophobia”; etc. – what do these terms mean, how have they been discussed in the literature, and why have they been included in the model? I realize the figure shows these relationships, but some discussion of each of the constructs in the model would be useful before launching into counts and study summaries. I would suggest that each construct needs its own mini-summary before launching into the count/study descriptions.
I’m also wondering if it would be worth adding a column to Table 2 indicating number of papers examining each component.
I would also suggest including a figure of the model showing the results of the review – i.e. those paths for which there is relatively strong evidence (gender, information on health and environment, lower price, meat attachment, food neophobia, lower situational appropriately of consuming non-meat protein), and those with less evidence. Or perhaps low, medium, high. This would provide a visual summary of the findings that I think would be useful for the reader. For example, perhaps thick, bold path lines could indicate more evidence, thinner lines for limited evidence, and dotted lines for no evidence/gaps.
Minor issues:
There are a few typos / syntax issues, etc.
Line 49: “Clearly, decreasing the portion size of meat would likely be more feasible.” Would it? Support for this statement?
Methods: did more than one researcher examine/read articles? Was there a process in place if they disagreed on whether to include or exclude? A bit more discussion of the process would strengthen an assertion of methodological rigor.
Lines 146-148 – I didn’t quite understand how this worked with surveys(?) Or were these all experimental studies?
Table 1 – there were inconsistencies in how studies were referred to; also please include a note, or write out, acronyms
Did all studies (whether surveys or experiments) specify the non-meat substitute? I found that interesting – I suppose I would have expected some, particularly surveys, may have just asked about meat substitutes in general.
Gender and Sex – these are different concepts. Did some studies use “gender” and others use “sex”? Which is more appropriate? Why the difference in different studies?
There are numerous places in the paper where the authors say something like, for example, “gender also varied between…” (lines 197-198) when I believe they actually mean that the impact of gender also varied between… Similarly, the authors state “factors influencing the replacement of…” (line 221) when I believe they actually mean “factors influencing the acceptance of replacement…”. (see also lines 235; 257, etc.) Please go back through the paper to correct.
Line 249 – which was preferred?
Line 325 – need to explain what “willingness to pay” experiments are
Lines 345-346 – always? More often?
Line 356 – define
Line 365- what is this? Why is it important?
Lines 682-683 – I didn’t understand this. Isn’t the review based on studies?
Lines 689-690 – “tremendous” is a loaded word – suggest deleting
Lines 761-765 – these other frameworks should have been discussed in the Introduction (see my earlier comment)
Round 2
Reviewer 2 Report
Thank you for addressing my concerns with this paper. I think it is a very interesting review that will increase the knowledge base in this area. There are still a few places where the grammar, syntax, or punctuation could be improved. However, overall it is well organised and written.
